# Programming mRNA decay to modulate synthetic circuit resource allocation

Ophelia S. Venturelli[1,2], Mika Tei[1,2], Stefan Bauer[3], Leanne Jade G. Chan[4], Christopher J. Petzold[4] & Adam P. Arkin[1,2,3,5]

Synthetic circuits embedded in host cells compete with cellular processes for limited intracellular resources. Here we show how funnelling of cellular resources, after global transcriptome degradation by the sequence-dependent endoribonuclease MazF, to a synthetic circuit can increase production. Target genes are protected from MazF activity by recoding the gene sequence to eliminate recognition sites, while preserving the amino acid sequence. The expression of a protected fluorescent reporter and flux of a high-value metabolite are significantly enhanced using this genome-scale control strategy. Proteomics measurements discover a host factor in need of protection to improve resource redistribution activity. A computational model demonstrates that the MazF mRNA-decay feedback loop enables proportional control of MazF in an optimal operating regime. Transcriptional profiling of MazF-induced cells elucidates the dynamic shifts in transcript abundance and discovers regulatory design elements. Altogether, our results suggest that manipulation of cellular resource allocation is a key control parameter for synthetic circuit design.

[1] California Institute for Quantitative Biosciences, University of California Berkeley, Berkeley, California 94158, USA. [2] Department of Bioengineering, University of California Berkeley, Berkeley, California 94720, USA. [3] Energy Biosciences Institute, University of California Berkeley, Berkeley, California 94704, USA. [4] Joint BioEnergy Institute and Biological Systems and Engineering Division, Lawrence Berkeley National Laboratory, Berkeley, California 94720, USA. [5] Environmental Genomics and Systems Biology, Lawrence Berkeley National Laboratory, Berkeley, California 94720, USA. Correspondence and requests for materials should be addressed to O.S.V. (email: venturelli@wisc.edu) or to A.P.A. (email: aparkin@lbl.gov).

Engineered biological systems have diverse applications in medicine, bioenergy and agriculture[1]. Novel cellular behaviours can be programmed by interacting networks of biomolecules to process information from the environment and execute target functions. These synthetic biomolecular circuits interact with endogenous cellular processes through competition over shared resources that include ribosomes, transfer RNAs (tRNAs), RNA polymerases, amino acids and nucleotides[2,3]. Resource utilization influences the predictability, function and evolutionary stability of engineered networks and constrains the achievable parameter space for synthetic circuit design[4].

Cells operate with a limited resource quota, which manifests as a trade-off in the partitioning of energy between cellular processes and synthetic circuit functions[1,3,5,6]. A core challenge is to rewire cellular regulation to optimally distribute resources between the host-cell and synthetic circuit processes. While there are numerous mechanisms to control target gene expression including engineered promoters[7], protein degradation[8] or CRISPRi[9–11], limited technologies exist to globally redistribute resources and reprogramme cellular state. Novel strategies should be developed to manipulate genome-wide gene expression patterns to optimize a target function.

RNA degradation more rapidly and efficiently redistributes ribosomes, a crucial limiting resource in Escherichia coli[6,12], compared to transcriptional control. Viruses capitalize on messenger RNA (mRNA) decay to reduce competition for the host cell translational machinery during developmental transitions and implement temporal gene expression programmes[13,14]. To exploit RNA decay for synthetic circuit resource redistribution in E. coli, we repurposed a sequence-specific ribonuclease MazF[15] whose recognition site 'ACA' is present in 96% of E. coli coding sequences. The MazF recognition site can be eliminated from the synthetic circuit while preserving the amino acid content, allowing cellular resources to be reallocated towards synthetic gene expression by eliminating competing processes.

Here we show that MazF activity induces a global cellular physiological shift that can be exploited to enhance synthetic circuit expression. These results suggest that the MazF resource allocator controllably redistributed core cellular subsystems to support a synthetic circuit and an engineered metabolic pathway. The former is further enhanced by protection of specific host-cell factors and use of the orthogonal RNA polymerase from T7 bacteriophage (T7 RNA polymerase) to transcribe genes in the synthetic circuit. Shotgun proteomics is used to identify a host factor in need of protection to prevent loss of translational efficiency following MazF induction. Our results demonstrate that the activity of the mRNA-decay feedback loop is a critical parameter for the resource allocator. A dynamic computational model of the circuit is constructed to interrogate the role of feedback on growth and circuit properties. Transcriptional profiling of MazF-induced cells is used to evaluate the correlation between the number of MazF sites and the impact of MazF expression on network activity. To pinpoint major parameters that influence MazF-induced decay rates, we examine the number and positioning of MazF recognition sites on the expression of a fluorescent reporter gene. In sum, these results suggest a platform for global manipulation of resource pools as a key parameter for modulating synthetic circuit behaviour.

## Results

**Characterization of MazF for resource allocator design.** To explore whether manipulation of resource allocation could predictably modulate circuit behaviour, we needed to develop a comprehensive reallocation mechanism that preserved core processes required for a target function, while downregulating competing pathways. MazF is a sequence-dependent and ribosome-independent endoribonuclease that cleaves the recognition site 'ACA' in single-stranded RNA[15,16]. Approximately 96% of E. coli coding sequences contain at least one MazF recognition site (Supplementary Fig. 1a). Thus, induction of MazF should inhibit cellular processes other than those protected from its action.

We characterized the impact of MazF on expression of a target gene mCherry that contained nine recognition sites in the coding sequence (mCherry-U) or was recorded to not contain any sites using alternative codons (mCherry-P). mazF was introduced into an intergenic genomic site under control of an aTc-inducible promoter ($P_{TET}$) in an E. coli strain deleted for mazF (strain S2 in Supplementary Table I). The total fluorescence of mCherry-P and mCherry-U were similar in the absence of MazF, indicating that recoding the transcript did not modify expression (Fig. 1b). The MazF induction ratio is a metric used to quantify resource redistribution activity, and is defined as the ratio of total mCherry-P fluorescence in the presence to absence of MazF. Following 10 h of induction with 0 or 5 ng ml$^{-1}$ aTc, the MazF induction ratio was <1 for mCherry-U and 5 for mCherry-P (Fig. 1c). The sequence protection ratio of total fluorescence, defined as the ratio of mCherry-P to mCherry-U, was ~1 or 19 in the absence or presence of MazF (Fig. 1d). Altogether, these data show that MazF significantly enhanced protected and inhibited unprotected gene expression.

To map the relationship between MazF expression and resource redistribution activity, growth and mCherry-X (X denotes U or P) expression were measured across a broad range of aTc concentrations. The total fluorescence of mCherry-U driven by an arabinose-inducible promoter ($P_{BAD}$) was reduced up to 4-fold in response to aTc (Supplementary Fig. 2). In the presence of aTc, the MazF induction ratio of total fluorescence was enhanced (Fig. 1e), whereas the total biomass was lower (Supplementary Fig. 3a). The MazF induction ratio of fluorescence divided by OD600 increased with aTc and arabinose (Supplementary Fig. 3b). While the biomass normalization factor altered the quantitative value of the induction ratio, the qualitative relationship between MazF activity and protected gene expression was unmodified (Fig. 1e and Supplementary Figs 2 and 3b). These data highlight that mCherry-P expression and biomass synthesis were inversely correlated in response to MazF. In sum, our results suggest that the enhancement of the protected gene mCherry-P in MazF-induced cells is due to augmented synthesis.

To interrogate the temporal variation in expression in MazF-induced cells, cell populations were induced with mCherry-P at three time points following exposure to MazF. To account for variability in biomass across conditions, we evaluated fluorescence divided by OD600 since the qualitative relationships were not altered by the biomass normalization factor (Fig. 1b,e; Supplementary Figs 2 and 3b). To compare expression across conditions, fluorescence divided by OD600 was normalized to the maximum expression level across all conditions following 12 h of induction with 5 ng ml$^{-1}$ aTc. In the absence of MazF, delayed induction by 2 h reduced mCherry-P expression by 85% (Supplementary Fig. 4a), whereas cells induced with MazF displayed a 34% decrease in mCherry-P expression (Supplementary Fig. 4b). These data indicate that heterologous expression was significantly attenuated by delayed induction in the absence of MazF, presumably by the transition from exponential to stationary phase. By contrast, delays in the induction of mCherry-P reduced expression by a smaller magnitude in the presence of MazF, indicating that MazF-induced cells preserved high-metabolic activity for a period of time.

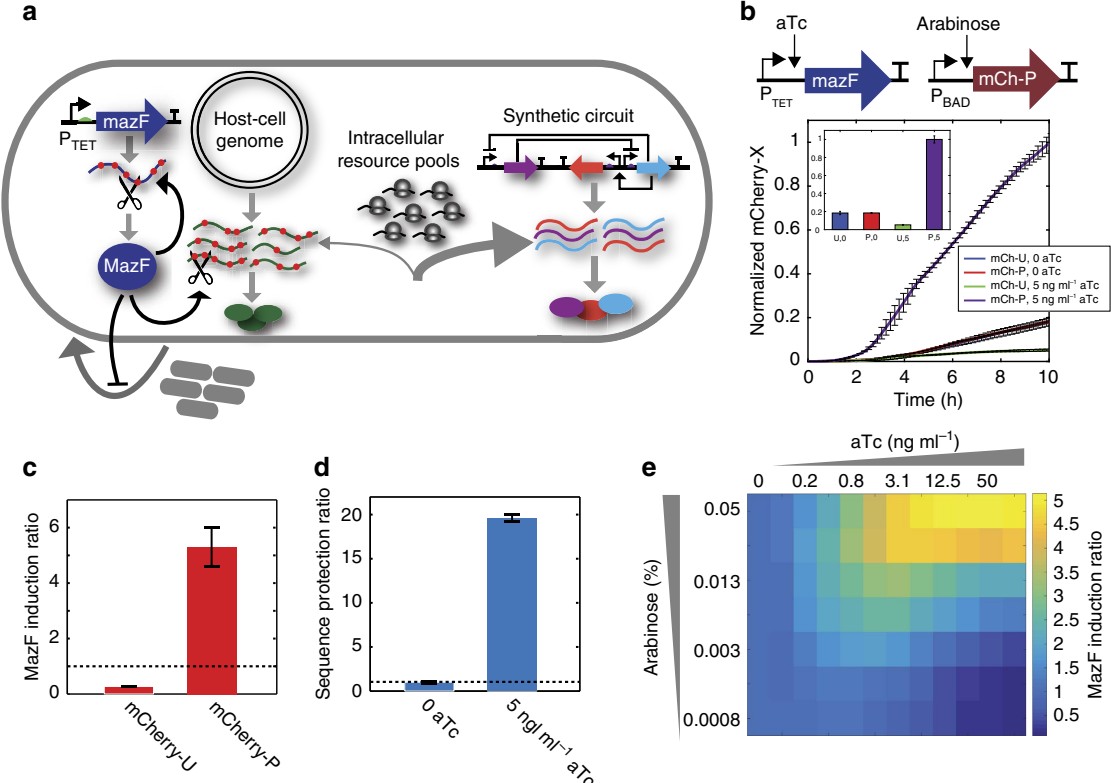

**Figure 1 | Redistributing resources in *E. coli* by programming mRNA decay. (a)** Schematic diagram of the MazF resource allocator. Host-cell transcripts containing MazF recognition sites ('ACA') are targeted for cleavage. The MazF site can be removed from target genes while preserving the amino acid sequence. As such, MazF down-regulates transcripts that compete with the protected synthetic circuit for limiting resources, yielding an increase in protected gene expression. **(b)** MazF and protected mCherry (mCherry-P) were controlled by an aTc and arabinose-inducible promoter (top), respectively. Time-series measurements of total fluorescence normalized to the maximum steady-state value ($t = 10$ h) across conditions for cells expressing unprotected mCherry (mCherry-U) or mCherry-P in the presence (5 ng ml$^{-1}$ aTc) or absence (0 ng ml$^{-1}$ aTc) of MazF. Cells were induced with 0.05% arabinose. Bar plot showing the steady-state normalized mCherry fluorescence (inset). **(c)** MazF induction ratio, defined as the total fluorescence of mCherry-X in the presence (5 ng ml$^{-1}$ aTc) to absence (0 ng ml$^{-1}$ aTc) of MazF. Cells were induced for 10 h with 0.05% arabinose. **(d)** Sequence protection ratio, defined as the total fluorescence ratio of mCherry-P to mCherry-U in the presence (5 ng ml$^{-1}$ aTc) or absence (0 ng ml$^{-1}$ aTc) of MazF. Cells were induced for 10 h with 0.05% arabinose. **(e)** Heat-map of MazF induction ratio of total fluorescence following 10 h of induction across a range of arabinose and aTc concentrations. Error bars represent 1 s.d. ($n = 3$).

To distinguish whether transcriptional or translation activity dominated the enhancement of mCherry-P in response to MazF, *mCherry-P* mRNA was measured using quantitative real-time PCR (qPCR). The *mCherry-P* mRNA fold change following 56 min of induction with 0 or 5 ng ml$^{-1}$ aTc relative to *mCherry-P* mRNA abundance at the beginning of the experiment ($t = 0$) was similar in the presence or absence of MazF (Supplementary Fig. 5). These data show that MazF did not significantly alter the *mCherry-P* transcription rate over this period of time. Therefore, these results suggest that MazF activity augmented the translation rate of *mCherry-P*.

**Enhancement of gluconate activity using MazF circuit.** The gluconate pathway competes directly with biomass synthesis by redirecting glucose into gluconate via glucose dehydrogenase (Gdh, Fig. 2a). To determine the impact of MazF on metabolic flux, biomass and gluconate were measured as a function of time (see Methods) in cells expressing protected Gdh (*gdh-P*) or unprotected Gdh containing 10 MazF recognition sites (*gdh-U*) controlled by a $P_{LAC}$ promoter. These experiments were conducted in a strain background that contained genetic modifications to inhibit gluconate metabolism and decouple glucose phosphorylation and transport to efficiently utilize glucose as a substrate for target metabolic pathways (strain S1 in Supplementary Table I)[17].

As expected, cell growth was inhibited by MazF induction whereas the uninduced population continued to grow as a function of time (Fig. 2b). Cells bearing *gdh-P* driven by a $P_{LAC}$ promoter displayed up to a three-fold higher gluconate concentration and five-fold higher gluconate per unit time in the presence of MazF compared to cells that were not induced with aTc (Fig. 2c; Supplementary Fig. 6a). The gluconate titre was 85% higher for cells induced with MazF compared to cells that were not induced following 18.25 h (Fig. 2d). A protected fluorescent reporter sfGFP (sfGFP-P) N-terminally fused to Gdh-U or Gdh-P increased up to 3.3 and five-fold as a function of aTc (Supplementary Fig. 6b). These data demonstrated that the MazF resource allocator could enhance metabolic flux by protecting genes in a target metabolic pathway.

**Protection of host-factors to enhance resource allocation.** Synthetic circuits depend on a dense network of host-genes including the transcriptional and translational machinery. Therefore, MazF-mediated decay of host factors could impact circuit functions. To investigate whether protection of support genes could improve the performance of the resource allocator, we tested whether protection of an orthogonal RNA polymerase T7 could enhance the circuit output. A protected (T7-P) or unprotected T7 RNA polymerase (T7-U containing 50 MazF

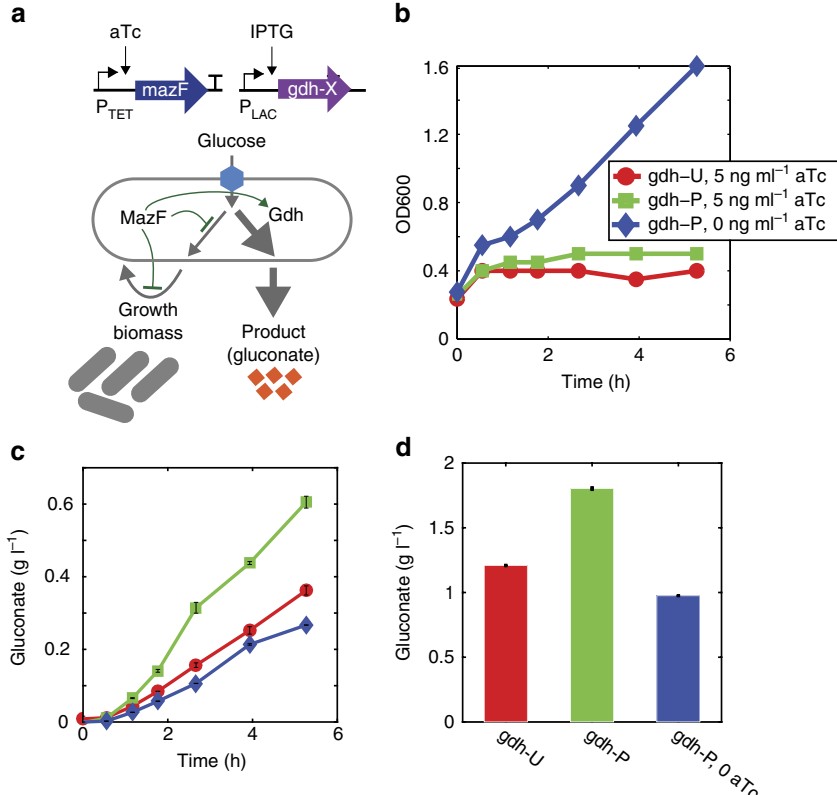

**Figure 2 | The MazF resource allocator enhanced gluconate production.** (**a**) Schematic diagram of the circuit design (top) and gluconate metabolic pathway (bottom). Glucose dehydrogenase (Gdh) transforms glucose into gluconate and competes directly with biomass synthesis. MazF and glucose dehydrogenase (gdh) were controlled by an aTc ($P_{TET}$) and IPTG-inducible ($P_{LAC}$) promoter, respectively. (**b**) OD600 as a function of time for cells expressing Gdh that contained 11 (Gdh-U) or 0 recognition sites (Gdh-P) in response to 5 or 0 ng ml$^{-1}$ aTc (below). All cultures were induced with 1 mM IPTG and supplemented with 1.5% glucose. (**c**) Gluconate titre as a function of time. (**d**) Gluconate titre following 18.25 h of induction. Error bars represent 1 s.d. from the mean of technical replicates ($n = 3$).

sites) controlled by an IPTG-inducible promoter ($P_{LAC}$) was used to drive the expression of mCherry (Fig. 3a). The combination of T7-P and mCherry-P yielded a 21 or 7.6-fold higher expression level of mCherry compared T7-P, mCherry-U or T7-U, mCherry-P in the presence of MazF (5 ng ml$^{-1}$ aTc) and 1 mM IPTG. T7-P regulating an N-terminal fluorescent protein fusion of mCherry-P to Gdh-P (mCherry-P-Gdh-P) exhibited a 1.4 and 15-fold higher expression compared to T7-P, mCherry-P-Gdh-U or T7-U, mCherry-P-Gdh-P (Supplementary Fig. 7). The mCherry expression level of the T7-X, mCherry-X (Fig. 3a) and T7-X, mCherry-X-Gdh-X (Supplementary Fig. 7) circuits were differentially enhanced by protection of T7 RNA polymerase or the reporter gene (*mCherry-X* or *mCherry-X-gdh-X*) in the presence of MazF. Thus, the quantitative value of the enhancement by protection of specific genes in a circuit depended on the circuit composition.

Defining translation factors in need of protection is challenging since the basic translation machinery consists of 78 factors including ribosomal proteins and aminoacyl-tRNA synthases[18]. To identify candidates, the proteome of MazF-induced cells was measured as a function of time. The majority of the proteome (216 measured proteins) and 91% of 35 detectable ribosomal proteins varied by <10% following 5 h of induction, demonstrating that highly abundant proteins were stable for hours following exposure to MazF (Supplementary Fig. 8a). Ribosomal protein subunits S9, S20 and L17 decreased by ∼20% and an essential elongation factor EF-Ts decreased by approximately 80% following 5 h of induction with MazF (Supplementary Fig. 8b). In the presence of MazF, a protected

version of EF-Ts (EF-Ts-P) driven by an IPTG-dependent promoter ($P_{LAC}$) significantly enhanced the expression of mCherry-P compared to cells that were not induced with EF-Ts-P (Fig. 3b). These results indicated that genome-wide measurements could be used to discover support genes in need of protection to augment resource redistribution activity.

Global mRNA decay could generate imbalances in the expression levels of genes in a regulatory network. For example, high concentrations of truncated mRNA fragments could saturate exonucleases that process these fragments into mono-nucleotides[19]. Further, mRNA cleavage generates ribosome stalling at the 3′ end of the mRNA, referred to as non-stop complexes, which require the action of ribosome recycling factors to rescue the ribosomes[20]. RNase R is a multifunctional protein that exhibits ribonuclease and ribosome recycling factor activities[21]. Co-expression of MazF and protected version of RNase R (RNase R-P) significantly enhanced the expression of mCherry-P compared to cells expressing only MazF (Fig. 3b). However, co-expression of EF-Ts-P and RNase R-P did not yield an additional enhancement in the level of mCherry-P in the presence of MazF compared to cells expressing either of the single support genes, RNase R or EF-Ts-P (Supplementary Fig. 9). These results suggested that epistasis among support genes could potentially limit incremental improvement of resource redistribution activity.

**Dissecting the role of the MazF mRNA-decay feedback loop.** The *mazF* transcript is enriched for recognition sites

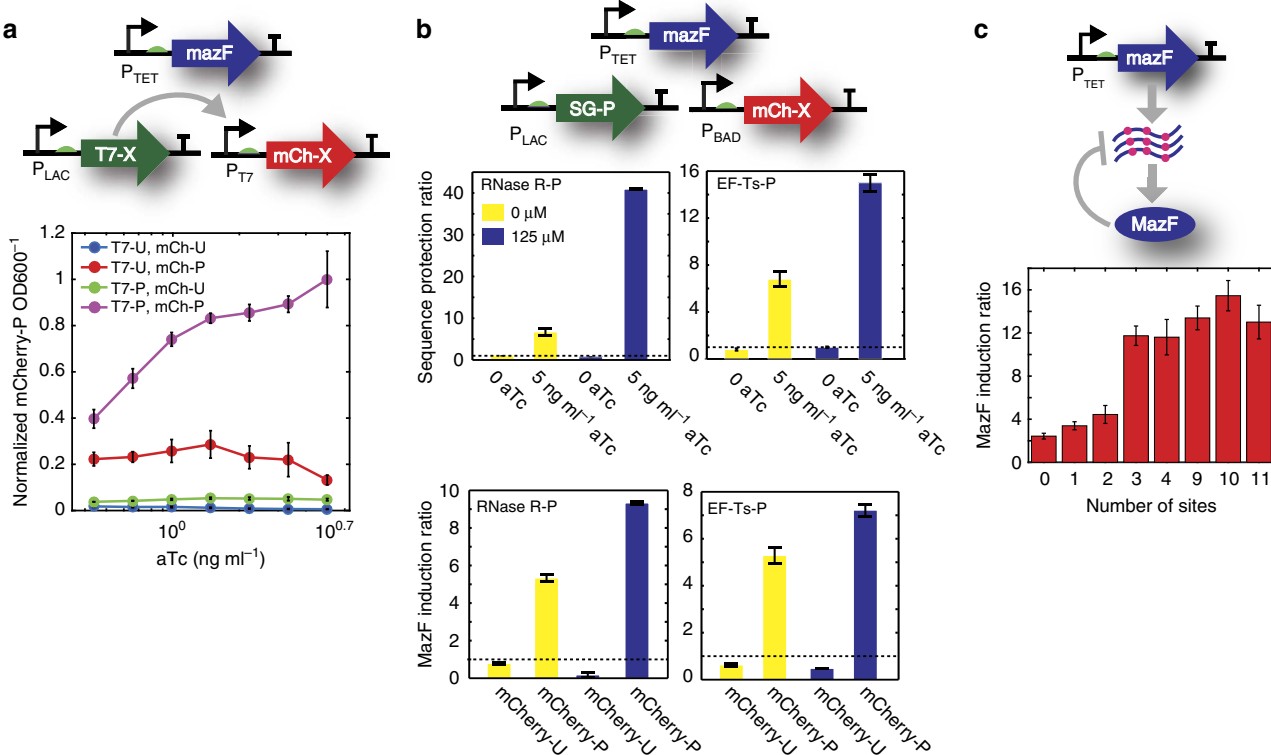

**Figure 3 | Improvement in resource redistribution activity via protection of key support genes and evaluation of the role of the MazF mRNA-decay negative feedback loop. (a)** Schematic of the orthogonal T7 RNA polymerase resource allocator circuit (top). MazF, T7 RNA polymerase (T7-X) and mCherry-X were controlled by an aTc ($P_{TET}$), IPTG ($P_{LAC}$) and T7 ($P_{T7}$) regulated promoter, respectively. Normalized fluorescence divided by OD600 as a function of aTc for cells expressing combinations of T7-U or T7-P and mCherry-U or mCherry-P following 8.3 h of induction with 1 mM IPTG (bottom). Error bars represent 1 s.d. ($n = 3$). **(b)** Schematic of support gene (SG-P) circuit (top). The support genes included protected host factors RNase R-P and EF-Ts-P. MazF, protected support genes and mCherry-X were controlled by $P_{TET}$, $P_{LAC}$ and $P_{BAD}$, respectively. We measured the expression of mCherry-P divided by OD600 in the presence and absence of MazF induction. The induction ratio is defined as the division of the former quantity by the latter. The sequence protection ratio is defined as the ratio of mCherry-P OD600$^{-1}$ to mCherry-U OD600$^{-1}$ in the presence or absence of MazF. Sequence protection ratio (middle) and MazF induction ratio (bottom) in the presence (5 ng ml$^{-1}$ aTc, 125 uM IPTG) or absence (0 ng ml$^{-1}$ aTc, 0 ng ml$^{-1}$ IPTG) of IPTG or aTc. Cells were induced with 0.05% arabinose for 8.3 h. Error bars represent 1 s.d. ($n = 4$). **(c)** Schematic of MazF mRNA-decay feedback loop (top). MazF induction ratio of fluorescence divided by OD600 for cells expressing *mazF* transcripts that varied in the number of recognition sites (P37-43 in Supplementary Table I). mCherry-P was regulated by a IPTG-inducible promoter ($P_{LAC}$). Cells were induced with 0 or 5 ng ml$^{-1}$ aTc and 1 mM IPTG for 9.2 h. Error bars represent 1 s.d. ($n = 4$).

(Supplementary Fig. 1b), establishing an mRNA-decay negative feedback loop. We suspected that protection of MazF could enhance circuit performance. However, the feedback loop may modulate the regulatory dynamics of MazF and therefore influence resource redistribution activity. To investigate this possibility, we probed the role of the mRNA-decay feedback in the MazF resource allocator.

Cells (strain S3 in Supplementary Table I) bearing *mazF-U* on a low copy plasmid (plasmid P1 in Supplementary Table I) controlled by an aTc-inducible promoter ($P_{TET}$) and induced with 5 ng ml$^{-1}$ aTc exhibited a lower steady-state *mazF* mRNA level compared to cells expressing *mazF-P* (Supplementary Fig. 10a), demonstrating that the feedback loop was actively regulating the abundance of the *mazF* transcript. Corroborating this result, a 35% lower threshold of aTc was required to inhibit growth in a strain expressing MazF-P compared to MazF-U (Supplementary Fig. 10b), suggesting that protection of *mazF* mRNA yielded a higher MazF protein level. The Hill coefficients of OD600 as a function of aTc following 11.2 h of induction were 2.6 and 5.9 for cells induced with MazF-U or MazF-P, revealing an ultrasensitive relationship between MazF activity and biomass synthesis that was significantly increased in the absence of the MazF mRNA-decay feedback.

Contrary to expectation, cells expressing MazF-U displayed significantly higher mCherry-P expression compared to cells expressing MazF-P across a broad range of aTc concentrations, highlighting that the negative feedback loop was a critical regulatory feature for the MazF resource allocator (Supplementary Fig. 10c). To further investigate the quantitative relationship between feedback loop strength and resource redistribution activity, we examined growth and protected reporter gene expression in cells (strain S3 in Supplementary Table I) bearing *mazF* sequences that varied in the number of recognition sites (Fig. 3c; Supplementary Fig. 11). The MazF induction ratio of fluorescence divided by OD600 increased with the number of sites and the wild-type *mazF* sequence (nine sites) generated nearly the highest output expression level (Fig. 3c). In sum, these results indicated that the activity of the mRNA-decay feedback loop was a tunable knob that could be used to modulate circuit performance.

A mechanistic computational model of cellular resource allocation was constructed to provide insight into the role of the mRNA-decay negative feedback loop on circuit behaviour (Supplementary Note). The dynamic model represented the mRNA and protein levels of key species involved in the MazF resource allocator (Supplementary Fig. 12), which compete for

limiting ribosome pools including ribosomes ($r$), unprotected proteome ($p$), MazF ($mazFp$) and a protected reporter gene ($FP$). The growth rate ($\lambda$) function was based on a previous coarse-grained mechanistic model of gene expression and growth[22]. A detailed description of the model and parameters are in Supplementary Note and Supplementary Tables II and III.

The relationship between the $mazF$ transcription rate $\alpha_f$ and the $FP$ translation rate ($k_{transFP} = k_{trans}[rFP]$) is non-monotonic (Supplementary Fig. 13a), indicating that there is an optimal expression level of MazF to maximize resource redistribution activity. The model shows that the strength of the feedback loop, represented by the dissociation constant of MazF dimer ($mazFpd$) to the $mazF$ transcript $m_f$ ($K_{Df} = k_{rff}\, k_f^{-1}$), is inversely correlated with the dose-response ultrasensitivity of total steady-state MazF concentration ($mazF_T = 2 \times [pf]_{ss} + 2 \times [rf]_{ss} + 2 \times [ff]_{ss} + 2 \times [fe]_{ss} + 2 \times [mazFpd]_{ss} + [mazFp]_{ss}$, where ss denotes steady-state) as a function of $\alpha_f$ (Fig. 4a,b). Molecular mechanisms that realize ultrasensitivity include MazF dimerization[23], molecular sequestration[24,25] of mRNAs by ribosomes[26] or positive feedback[27]. In addition, thresholded control of $\lambda$ by $mazF_T$, which was observed in our experimental and modelling data (Fig. 4d; Supplementary Fig. 10b), could contribute to ultrasensitivity in the network. For high $K_{Df}$ corresponding to the open loop system, the model exhibits bistability manifesting as two stable steady states across a range of $\alpha_f$ values (Supplementary Fig. 13b). Since $m_p$ and $m_r$ compete for limiting ribosome pools (Supplementary Fig. 13c), bistability could arise via positive feedback[25] established by an increase in

the synthesis rate of $r$ as a consequence of MazF-dependent $m_p$ decay. The MazF mRNA-decay negative feedback loop enables proportional adjustment of the $mazF_T$[28] and reduces the potential for bistability by abolishing ultrasensitivity[25,29] (Fig. 4b). As such, $mazF_T$ concentration could be tuned to operate in the regime that maximized resource redistribution activity.

For a fixed value of $\alpha_f$, $k_{transFP}$ is inversely related to $K_{Df}$ (Fig. 4c), qualitatively recapitulating the increase in mCherry-P with the number of binding sites in the $mazF$ transcript (Fig. 3c). $\lambda$ and the total concentration of the unprotected gene $p$ decrease as a function of $\alpha_f$, mirroring the experimental data that showed lower OD600 and mCherry-U in the presence of aTc (Supplementary Figs 2,3a and 10b; Fig. 4d,e). The increase in ultrasensitivity of the dose response of $mazF_T$ versus $\lambda$ as a function of $K_{Df}$ (Fig. 4d) qualitatively reflected the enhanced ultrasensitivity of the steady-state dose response of aTc versus biomass (OD600) for cells expressing MazF-P compared to MazF-U (Supplementary Fig. 10b). The negative feedback loop strength is inversely related to the range of $\alpha_f$ values that enhance total steady-state $r$ concentration ($r_T$, Fig. 4f). Above a threshold value of $K_{Df}$, $r_T$ decreases monotonically with $\alpha_f$. The mRNA-decay negative feedback has important implications for resource allocator design by enabling precise tuning of the MazF operating point by establishing a proportional relationship between $\alpha_f$ and $mazF_T$. Indeed, this negative feedback may provide an evolutionary advantage for cells by preventing the deleterious effects of MazF overexpression that accelerated cell death (Supplementary Fig. 14).

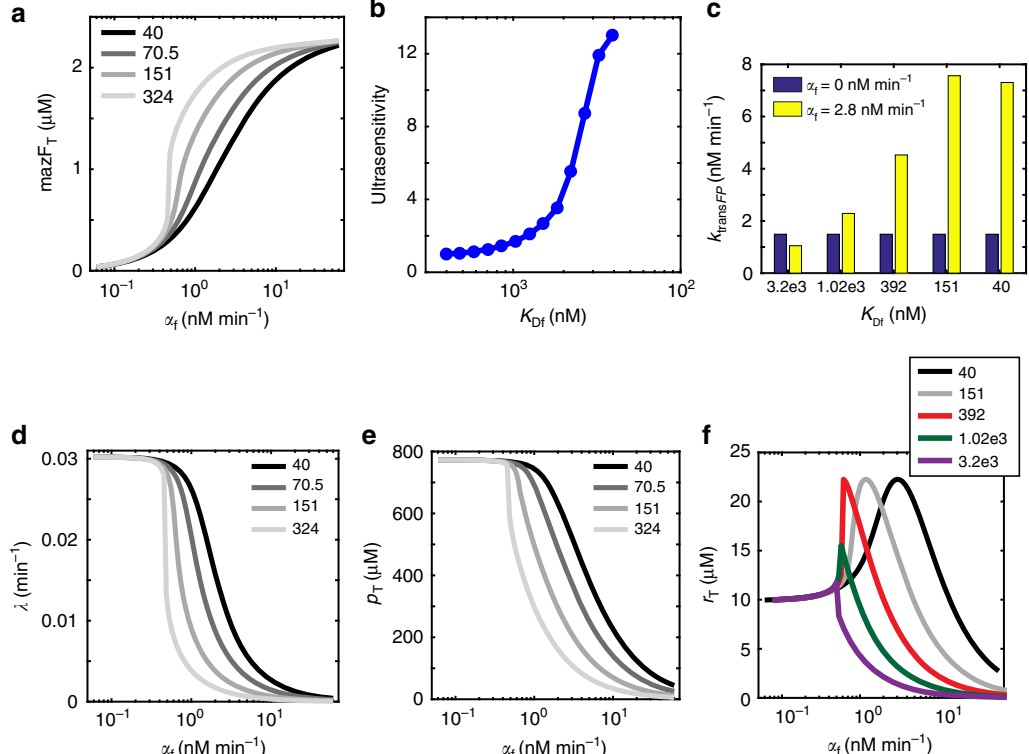

**Figure 4 | Probing the role of the MazF negative feedback loop in a dynamic computational model of resource allocation.** This model demonstrates that the MazF mRNA-decay feedback loop established proportional control of MazF in the absence of MazE ($\alpha_e = 0$). (**a**) Total MazF concentration at steady-state ($mazF_T$, $t = 278$ h) as a function of the transcription rate of $mazF$ ($\alpha_f$) across a range of dissociation constants ($K_{Df}$) in units of nM of MazF to $mazF$ mRNA ($m_f$). Here, $mazF_T = 2 \times [pf]_{ss} + 2 \times [rf]_{ss} + 2 \times [ff]_{ss} + 2 \times [fe]_{ss} + 2 \times [mazFpd]_{ss} + [mazFp]_{ss}$, where ss denotes steady-state. (**b**) Maximum logarithmic sensitivity (ultrasensitivity) of the dose response of $\alpha_f$ versus $mazF_T$ across a range of $K_{Df}$ values. (**c**) Steady-state translation rate of a protected gene $FP$ ($k_{transFP} = k_{trans}[rFP]_{ss}$) as a function of $K_{Df}$ in the presence ($\alpha_f = 2.8$ nM min$^{-1}$) or absence ($\alpha_f = 0$ nM min$^{-1}$) of MazF. (**d**) Steady-state growth rate ($\lambda$) as a function of $\alpha_f$ for different values of $K_{Df}$. (**e**) Steady-state total unprotected proteome ($p_T$) concentration as a function of $\alpha_f$ for different values of $K_{Df}$. (**f**) Steady-state total ribosome concentration ($r_T$) as a function of $\alpha_f$ for different values of $K_{Df}$.

**MazE feedback loop impacts growth and circuit properties**. Transcriptional profiling and proteomics measurements of MazF-induced cells (strain S2 in Supplementary Table I) revealed that the *mazE* transcript (Supplementary Fig. 15a) and MazE protein (Supplementary Fig. 15b) were up-regulated by aTc administration. MazE is a stoichiometric inhibitor of MazF activity by sequestering MazF into an inactive complex[30]. Stimulation of MazE synthesis in response to MazF activity establishes a molecular sequestration negative feedback loop. The protein abundance of MazF significantly exceeded MazE, explaining the lack of MazF inhibition in these conditions (Supplementary Fig. 15b). Since MazE could be used to control the activity of the MazF resource allocator, we examined the impact of MazE activity on growth and circuit properties in the model.

The transcription rate of *mazE* was a function of active MazF (mazFpd) in the model to capture the coupling between MazF induction and MazE synthesis (Supplementary Note). Increasing the maximum mazE transcription rate $\alpha_e$ reduced the total active MazF concentration (total active MazF concentration was defined as $[pf]_{ss} + [rf]_{ss} + [ff]_{ss} + [fe]_{ss} + [mazFpd]_{ss}$, where ss denotes steady-state; Supplementary Fig. 16a). As a result, a higher $\alpha_f$ was required to fully inhibit cell growth in the presence of MazE (Supplementary Fig. 16b). Increasing $\alpha_e$ shifted the regime of maximum resource redistribution activity towards higher $\alpha_f$ values (Supplementary Fig. 16c). Ultrasensitivity in the steady-state dose response of $\alpha_f$ versus total MazF (mazF$_T$ = $2 \times [pf]_{ss} + 2 \times [rf]_{ss} + 2 \times [ff]_{ss} + 2 \times [fe]_{ss} + 2 \times [mazFpd]_{ss} + 2 \times [cef]_{ss} + [mazFp]_{ss}$) was moderately enhanced by up to $\sim$23% in a narrow parameter regime corresponding to high $K_{Df}$ and intermediate $\alpha_e$ values, presumably via molecular sequestration (Supplementary Fig. 16d)[24]. However, ultrasensitivity was significantly reduced across a broad range of $\alpha_e$ values. The range of $\alpha_f$ that mapped to high resource distribution activity could be adjusted by modulating both the MazE and MazF mRNA-decay feedback loops. However, in contrast to the mRNA-decay feedback, increasing the strength of the MazE feedback moderately reduced the parameter range that mapped to optimal circuit performance (Supplementary Fig. 16c). In sum, MazE is a key control parameter for the MazF resource allocator that can be used to rapidly modulate growth and resource redistribution activity[31].

**Transcriptional profiling of MazF-induced cells**. To evaluate the genome-wide variation in transcript abundance following MazF exposure, RNA-seq measurements of MazF-induced cells were collected every 2 min for a total of 8 min using strain S2 induced with 5 ng ml$^{-1}$ aTc (Supplementary Table I). The majority of the 192 endogenous protected genes increased or remained constant following induction with MazF for 8 min (Fig. 5a). A balance between synthesis and decay catalysed by RNases and MazF determines transcript abundance. Therefore, it is challenging to directly decipher the MazF-dependent transcript decay rates. Nevertheless, the number of MazF sites was negatively correlated with the mean log2 fold change of transcript abundance following 8 min of induction with aTc, indicating that on average the number of MazF sites predicted the fold change across the transcriptome (Fig. 5b, Supplementary Fig. 17).

Partitioning the transcriptome fold change dynamics into three clusters (see Methods) revealed three temporal patterns in transcript abundance in response to MazF induction: down-regulation (K1, 460 genes), pulsatile response characterized by an increase in transcript abundance at early times and decrease following a delay (K2, 148 genes) or up-regulation (K3, 331 genes, Fig. 5c). We evaluated functional or regulatory enrichments ($P < 0.05$ using the Fisher's exact test) in each cluster to provide

insights into the physiological impact of MazF exposure (Supplementary Table V). Cell envelope and genes regulated by Fur, MraZ and LexA were enriched in the K1 cluster (Fig. 5c; Supplementary Fig. 18). MraZ is a transcriptional repressor that controls many genes involved in cell division and cell wall biosynthesis[32]. In addition, the cell division regulator *minE* mRNA decreased significantly in the RNA-seq data (Fig. 5a), corroborating a link between MazF activity and inhibition of cell division[33,34]. The K2 cluster was enriched for genes regulated by NikR, GlpR, GcvA, IHF, IscR and RstA and amino acid and anaerobic metabolism (Supplementary Fig. 18). K2 contained numerous regulatory categories (Supplementary Table V), suggesting that the pulsatile transcript dynamics could be established by an early increase in synthesis rates and delayed down-regulation due to mRNA-decay at a threshold concentration of MazF. Genes regulated by ArgR were enriched in the up-regulated cluster K3. In addition, 11 TCA cycle enzymes were up-regulated in the RNA-seq data ($P = 0.051$ enrichment in K3), suggesting that MazF-induced cells exhibited high metabolic activity (Supplementary Fig. 19; Supplementary Table V). Previous work has demonstrated that fumarate production increased the frequency of persister cells following antibiotic exposure[35]. A closer examination of the catabolic pathway revealed that fumarate producing enzymes were significantly induced, illustrating a connection between MazF activity and persistence via enhancement of fumarate flux[36,37] (Supplementary Fig. 19).

Cold-shock genes are selectively expressed in response to cold stress and perform diverse functions including unwinding of RNA secondary structures, modulation of ribosome and DNA/RNA chaperone activity[38]. The transcriptional profiling data revealed significant shifts in cold-shock *cspBCEFG* and associated *rbfA*, *rhlB*, *rhlE* and *deaD* transcript abundance as a function of time (Supplementary Fig. 20). IF-3, one of the major translation factors in *E. coli*, has been shown to mediate cold shock translational bias in response to cold stress[39,40]. IF-3 increased over four-fold in the proteomics data (Supplementary Fig. 8b) following 5 h of MazF induction, whereas the abundance of *infC* mRNA did not change significantly in response to MazF activity (Fig. 5a). Future work should interrogate the molecular mechanisms and functional connection among MazF activity, up-regulation of IF-3, and significant shifts in cold-shock transcript abundance.

As cold-shock transcripts were up-regulated in response to MazF activity, these sequences were promising candidates for engineering MazF-responsive promoters. To test the modularity of cold-shock induction by MazF, we constructed a tandem promoter composed of P$_{LAC}$ upstream of the *cspB* or *cspG* promoter, UTR and the first 14 amino acids of CspG or CspB N-terminally fused to sfGFP-P (Supplementary Fig. 21). MazF induction increased sfGFP-P by a maximum of 20 or 80-fold, demonstrating that the *cspB* and *cspG* regulatory sequences are modular control elements that directly respond to MazF activity as an input.

**Interrogation of parameters that impact MazF activity**. A quantitative understanding of the mapping between MazF site placement and cleavage efficiency could enable tuning of the timing and degrees of protection to inform resource allocator design. Previous work demonstrated that MazF activity was inhibited by strong secondary structures and ribosomes enhanced cleavage efficiency by unwinding mRNA secondary structures during translation[41]. To explore the dominant parameters that influence MazF cleavage efficiency, we varied the number and position of MazF recognition sites in the *mCherry* transcript

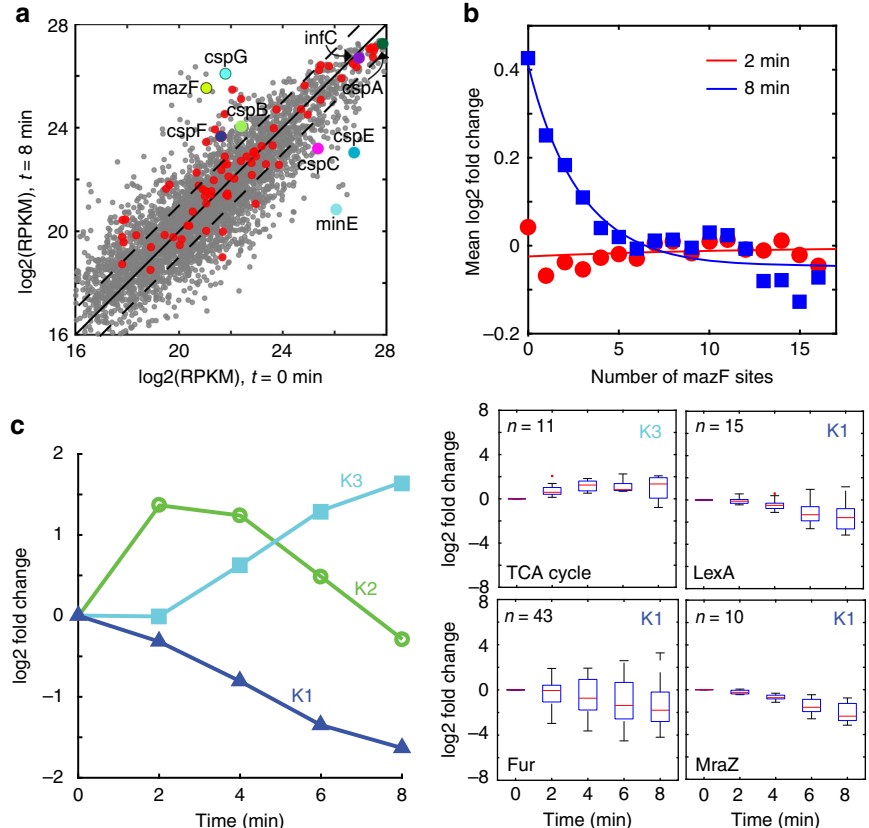

**Figure 5 | Time-series RNA-seq measurements of MazF-induced cells.** The mean RPKM value ($n = 2$) was log2 transformed. (**a**) Scatter plot of log2 transformed RPKM measurements before induction with MazF versus induction with MazF (5 ng ml$^{-1}$ aTc) for 8 min. Grey and red data points denote unprotected or protected transcripts larger than 80 nucleotides, respectively. Dashed lines represent a two-fold threshold in transcript abundance. *cspABCGEF*, *mazF* and *minE* transcripts are highlighted. (**b**) Scatter plot of the number of *mazF* sites for each gene versus mean log2 fold change following induction with 5 ng ml$^{-1}$ aTc for 2 or 8 min. A 5-point moving average was applied to the data. Lines represent fitted exponential functions to the data. (**c**) K-means clustering of log2 fold change of 939 genes (left) that exhibited correlated dynamics between biological replicates. Box plots (right) of representative functional or regulatory enrichments in the K1 and K3 clusters according to the Fisher's exact test ($P < 0.05$). On each box, the red line indicates the median, the bottom and top edges represent the 25th and 75th percentiles and '$+$' denote outlier data points. $n$ represents the number of genes in each category (Supplementary Table V).

(plasmids P21-36) in the S2 background strain (Supplementary Table I).

To map the relationship between position and cleavage efficiency, a single MazF site was inserted at 14 positions in *mCherry-P* (Supplementary Fig. 22). These *mCherry* sequences exhibited a broad range of expression levels in response to MazF (Supplementary Fig. 22a). The output was correlated with the predicted secondary structure Gibbs free energy ($\Delta G$) 38–47 bp upstream of the recognition site ($\rho$ ranged between $-0.7$ to $-0.5$, $P < 0.05$ using the Student's $t$-test) computed using NUPACK (Supplementary Fig. 22b,c). For sequences spanning upstream and downstream of the MazF site, mCherry expression was correlated ($\rho = -0.6$, $P < 0.05$ using the Student's $t$-test) with $\Delta G$ (39–40 bp, Supplementary Fig. 22d). However, the $\Delta G$ of the sequence downstream of the recognition site was not correlated with the expression level of mCherry across a broad range of window sizes (Supplementary Fig. 22e). Therefore, MazF cleavage efficiency could be predicted using the folding energy of the local mRNA secondary structure upstream or across the recognition site.

To provide insight into the programmability of MazF cleavage efficiency, we interrogated whether measurements of *mCherry* variants containing a single MazF site (Supplementary Fig. 22a) could predict the expression of *mCherry* sequences containing combinations of sites. mCherry expression decreased as a

function of the number of recognition sites in the presence of MazF (Supplementary Fig. 22f). The product of the single site *mCherry* expression levels could predict the expression of the multi-site variants ($P < 4e{-}6$ using the Student's $t$-test), suggesting that combinations of MazF recognition sites could be used to modulate the degree of transcript protection.

## Discussion

A major goal of synthetic biology and metabolic engineering is to develop strategies to control the resource economy of cells for switching between modes of growth and production[42]. During a production phase, cellular energy and resources are focused on specific pathways, while minimizing resource expenditure towards nonessential cellular operations. Towards these objectives, previous work leveraged tunable enzymatic degradation of a metabolic hub that determines the direction of metabolic flux to augment the yield and titre of a metabolic pathway two-fold[43]. While this strategy provided localized control of metabolic flux, it does not modulate the global allocation of subsystems such as transcription and translation. On a larger scale, inducible regulation of RNA polymerase subunits was recently used to control *E. coli* growth[35]. However, this mechanism cannot be generally applied to redirect resources towards engineered networks.

Here, we showed that synthetic circuits could exploit shifts in cellular physiological state due to MazF activity, suggesting that intracellular resources could be diverted via programmable mRNA decay. This approach could be harnessed for diverse applications by protecting genes in an engineered network and systematically discovering key support factors beyond the engineered pathway in need of protection. Recent advancements in DNA synthesis technologies will facilitate large-scale recoding of support genes to protect from MazF activity. A utility of this approach is to enhance target functions that compete directly with biomass synthesis, such as exploiting microbes as 'cell factories' to synthesize chemicals or biomolecules of interest. Further, MazF activity could potentially minimize unintended environmental impact due to cell proliferation, while allowing engineered cells to carry out a desired function in a complex environment. Coupling this strategy to dynamic regulation of MazE would enable periodic resuscitation of cellular sub-systems and maintain metabolic activity over longer time scales.

MazF regulates orders of magnitude more genes simultaneously compared to other technologies such as CRISPRi[45,46]. Homologues of MazF that recognize 3, 5 and 7-bp recognition sites have been identified in diverse bacterial species[47–49]. Active site mutations have been shown to modify the MazF sequence specificity, suggesting that protein engineering could be used to expand the diversity of MazF recognition site sequences[50]. The variation in recognition sequence specificity could be used to tune the number of genes targeted by MazF.

In addition to the unknown myriad effects of MazF-induction on network activities, there are several limitations to optimizing the MazF resource allocator. MazF activity increased the abundance of a set of host-cell transcripts (cluster K3 in Fig. 5c), which sequesters resources away from engineered circuits. However, this activation programme could be exploited by repurposing regulatory elements that respond to MazF activity to expand the resource allocator design. In addition, MazF activity has been shown to yield a heterogeneous ribosome pool by targeting a specific site of the 16S rRNA[51], which could manifest as translation bias for selected transcripts[52]. Decay of the unprotected proteome occurs on the time scale of hours, thus limiting the time scale required to shift metabolic flux. To rapidly manipulate metabolic flux, induction of MazF could be coupled with proteases[8] for targeted control of protein abundance. As the proteome decays, stoichiometric relationships required for protein activity must be maintained[31]. Further, MazF has been shown to establish a futile cycle of continuous RNA synthesis and decay, resulting in energy dissipation[36]. To minimize an energy deficit, orthogonal T7-P could be used to drive the engineered pathway, while at the same time inactivating native RNA polymerases.

Cells have evolved numerous feedback mechanisms to optimize ribosome concentrations to match changes in environmental conditions, including nutrient quality and abundance[53,54]. These growth-rate dependent couplings to cellular processes including transcription, translation and replication can influence the behaviour of synthetic circuits. In MazF-induced cells, the consequences of growth rate inhibition on cellular sub-systems remain unresolved. The stringent response is not activated in MazF-induced cells, which allows cells to maintain ribosome synthesis and cellular maintainence[55]. A detailed understanding of network activities and resource partitioning in MazF-induced cells will allow for exploitation of this unique physiological state for diverse biotechnological applications.

Top-down approaches such as MazF could be used to discover host factors that preserve high metabolic activity in the absence of growth. Genome engineering could be used to protect these pathways from MazF activity[56]. Optimal regulatory strategies should be designed to balance enhancement of resource redistribution activity and degradation of cellular support subsystems over long time scales. For example, MazF could be transiently induced until energy degrades to a threshold that triggers rapid inhibition of MazF activity via MazE and allows rebalancing of the proteome[57]. Altogether, advances in regulatory control strategies and large-scale recoding may enable the design of protected and unprotected orthogonal sub-genomes that dynamically switch between cellular operations.

## Methods

**Cloning and strain construction.** *mazF* was deleted from the *E. coli* BW25113 strain using lambda-red recombination and verified by colony PCR. MazF was introduced into an intergenic region referred to as SafeSite 1 (chromosomal position 34715) under control of an aTc-inducible promoter ($P_{TET}$). PCR amplifications were performed using Phusion High-Fidelity DNA Polymerase (NEB) and oligonucleotides for cloning and strain construction were obtained from Integrated DNA Technologies. Standard cloning methods were used to construct plasmids. Plasmids were derived from previously generated construct library[58]. A list of plasmids and strains used in this study can be found in Supplementary Table I.

**Growth conditions and plate reader experiments.** For plate reader experiments, cells were grown at 37 °C for ~6–8 h, and then diluted to OD600 of 0.01 in a 96-well plate (Corning) in LB Lennox media (Sigma). In 96-well plates, cells were grown in 200 µl volumes at 37 °C covered by a gas-permeable seal (Fisher Scientific) in a M1000 (Tecan) or Synergy 2 (BioTek) plate reader. Cells were cultured for 1–2 h in the plate reader before inducer administration. The method measured cell density (OD600) and fluorescence every 10 min for 15 h. The M1000 excitation and emission wavelengths were 485, 510 nm for GFP and 587, 610 nm for RFP (5 nm bandwidth). The BioTek excitation and emission wavelengths were 485, 528 nm for GFP and 560, 620 for RFP (20 nm bandwidth). The M1000 and Synergy 2 measured absorbance at 600 nm (OD600) to quantify total biomass. For each experiment, the minimum value of fluorescence or OD600 across all conditions was subtracted from fluorescence or OD600 measurements. Normalized fluorescence was computed by dividing by the maximum value across conditions. Normalized fluorescence divided by total biomass (OD600) was computed by dividing total fluorescence by OD600 and then normalizing to the maximum value across conditions. For plate reader experiments, biological replicates consisted of cells inoculated into different wells in a 96-well plate that were exposed to equivalent inducer concentrations.

**qPCR measurements.** Oligonucleotides for quantitative real-time PCR (sequences are listed in Supplementary Table IV) were designed using Integrated DNA Technologies. Total RNA of 500 ng was reverse transcribed using the iScript complementary DNA (cDNA) synthesis kit (Bio-Rad). The reaction mix contained 5 µl of SsoAdvanced Universal Probes Supermix (Bio-Rad), 0.5 µl primer and probe corresponding to 250 nM primers and 125 nM probe (20 × stock) and 0.5 µl of cDNA. qPCR measurements were performed using a CFX96 real-time PCR machine (Bio-Rad). The relative expression levels were determined by a $2^{-\Delta\Delta G}$ method. Each sample was normalized by the cycle threshold geometric mean using reference genes *rrsA* and *cysG*[59]. Biological replicates consisted of three *E. coli* cultures exposed to equivalent inducer concentrations (0 or 5 ng ml$^{-1}$). Three qPCR technical replicates were performed and averaged for each sample.

**Gluconate measurements.** KTS022IG mazF::Δ (strain S1 in Supplementary Table I) strains bearing pBbA6c-gdh-X (plasmid P8-9 in Supplementary Table I) and pBbS2k-mazF-U (plasmid P1) were grown in LB medium at 37 °C overnight and used to inoculate a 10 ml culture the next morning at an OD600 of 0.05. At OD600 of 0.3, 1.5% glucose, 1 mM IPTG and 5 or 0 ng ml$^{-1}$ were administered to the cultures. 1 ml samples were collected at the specified times and centrifuged at 5,000g for 5 min to isolate the supernatant. The supernatant samples were analysed for gluconic acid using a 1,200 Series liquid chromatography system (Agilent Technologies) coupled to an LTQ-XL ion trap mass spectrometer (Thermo Scientific) equipped with an electrospray ionization source. Aliquots of the diluted samples were injected onto a Rezex ROA-Organic Acid H+ (8%) (150 mm × 4.6 mm) column (Phenomenex) equipped with a Carbo-H+ (4 × 3 mm$^2$) guard column (Phenomenex). Gluconic acid was eluted at 55 °C at ~3.5 min with an isocratic flow rate of 0.3 ml min$^{-1}$ of 0.5% (v/v) formic acid in water. Precursor ion *m/z* 195.1 was selected in negative ion mode using an isolation window of *m/z* 2 and was fragmented with a normalized collision energy of 35. Fragment ions were analysed in the range of *m/z* 50–200. *m/z* 128.5–129.5 was selected as pseudo-MRM transition for compound quantification. Resulting peak areas were compared to an external standard calibration in the range of 0.2–200 uM. The source parameters were ion spray voltage: 4 kV; capillary temperature: 350 °C; capillary voltage: −2 V; tube lens voltage: −40 V; sheath gas flow: 60; auxiliary gas flow: 5; and sweep gas flow: 10 (all arbitrary units). Technical replicates were performed by measuring the sample three independent times.

The experiment was repeated three independent times. These experiments showed that the MazF-induced cells expressing Gdh-P yielded the highest gluconate concentrations compared to uninduced cells and MazF-induced cells expressing Gdh-U.

**Proteomics.** BW25113 mazF::$\Delta$, SafeSite1::tetR-P$_{TET}$-mazF (strain S2 in Supplementary Table I) was grown overnight in LB at 37 °C and then diluted to an OD600 of 0.05 in a 500 ml LB culture. At OD600 of 0.5, cell populations were induced with 5 ng ml$^{-1}$ aTc and 40 ml of the cultures were collected approximately every hour and centrifuged for 5 min at 4,300$g$. Proteomic samples were prepared for analysis by lysing the cell pellets and extracting the proteins using the chloroform/methanol precipitation method[60]. The proteins were resuspended in 100 mM AMBIC with 20% methanol and reduced with tris(2-carboxyethyl) phosphine for 30 min, followed by addition of iodoacetamide (IAA; final conc. 10 mM) for 30 min in the dark, and then digested overnight with MS-grade trypsin (1:50 w/w trypsin: protein) at 37 °C. Peptides were stored at $-20$ °C until analysis.

Samples were analysed on an Agilent 1290 UHPLC—6550 QTOF liquid chromatography mass spectrometer (LC–MS/MS; Agilent Technologies) system and the operating parameters for the LC–MS/MS system were described previously[60]. Peptides were separated on a Sigma-Aldrich Ascentis Express Peptide ES-C18 column (2.1 × 100 mm$^2$, 2.7 mm particle size, operated at 60 °C) and a flow rate of 0.4 ml min$^{-1}$. The chromatography gradient conditions were as follows: from the initial starting condition (98% buffer A containing 100% water, 0.1% formic acid and 2% buffer B composed of 100% acetonitrile, 0.1% formic acid) the buffer B composition was held for 2 min then increased to 10% over 3 min; then buffer B was increased to 40% over 117 min, then increased to 90% B over 3 min and held for 8 min, followed by a ramp back down to 2% B over 1 min, where it was held for 6 min to re-equilibrate the column to the original conditions. The data were analysed with the Mascot search engine version 2.3.02 (Matrix Science) and filtered and validated using Scaffold v4.3.0 (Proteome Software Inc.)[60]. Replicates consisted of four aliquots of an *E. coli* culture exposed to 5 ng ml$^{-1}$ aTc for different lengths of time. Shotgun proteomics was performed independently on each sample.

**RNA-seq library construction and sequencing.** BW25113 mazF::$\Delta$, Safe-Site1::tetR-P$_{TET}$-mazF (strain S2 in Supplementary Table I) was grown overnight in LB at 37 °C and then diluted to an OD600 of 0.05 in a 10 ml LB culture. At an OD600 of 0.5, cells were induced with 5 ng ml$^{-1}$ aTc. Samples were collected as follows: 200 μl of the cell cultures were added to 400 μl of RNAprotect (Qiagen) to stabilize the RNA, incubated for 5 min at room temperature and then spun down for 10 min at 5,000$g$. Total RNA was isolated using RNeasy purification kit and treated with DNAase I (Qiagen). The Functional Genomics Lab (FGL), a QB3-Berkeley Core Research Facility at UC Berkeley, constructed the sequencing libraries. At the FGL, Ribo-Zero rRNA Removal Kits (Illumina) were used to remove ribosomal RNA and ERCC RNA Spike-In Control Mixes (Ambion by Life Technologies) were added to the samples. The library preparation was performed on an Apollo 324 with PrepX RNAseq Library Prep Kits (WaferGen Biosystems, Fremont, CA), and 18 cycles of PCR amplification was used for index addition and library fragment enrichment. Biological replicates consisting of two *E. coli* culture aliquots exposed to 5 ng ml$^{-1}$ aTc were collected at the specified times. RNA-seq libraries were constructed independently from each sample.

**RNA-seq data analysis.** The read counts were mapped onto the MG1655 genome using Bowtie 1 (ref. 61) on the galaxy webserver[62]. Reads per kilobase of transcript per million (RPKM) was computed by multiplying the number of mapped reads by 10$^9$ and then dividing by the gene length and median number of total reads for each condition. For clustering analysis, the correlation coefficient ($\rho = 0.9$) between two biological replicates as a function of time was used as a threshold to remove genes that exhibited variability between replicates. The log2 fold change was partitioned into clusters using the K-means algorithm (MATLAB). To determine an optimal number of clusters, the sum of squared errors was computed for each data point from the corresponding cluster centroid across a range of K-values (1–10). The Elbow method was used as a heuristic to select the optimal number of partitions that minimizes the sum of squared errors. The Fisher's exact test ($P < 0.05$) was used to evaluate enrichment of genes based on TIGRFAM annotation (MicrobesOnline) or transcription factor network (RegulonDB). Supplementary Table V contains a list of genes in the enriched categories.

**Computational modelling.** We used custom code for computational modelling and data analysis in MATLAB (Mathworks) and Python. Details about the model construction are provided in Supplementary Note. Model species and parameters are described in Supplementary Tables II and III. Supplementary Software contains MATLAB code for the MazF resource allocation model.

**Characterization of cell viability.** A BW25113 mazF::$\Delta$ strain (strain S3 in Supplementary Table I) transformed with pBbS2k-mazF-U or pBbS2k-mazF-P (plasmid P1-2 in Supplementary Table I) was grown overnight at 37 °C in LB media and then diluted to an OD600 of 0.01 in 5 ml LB media. At an OD600 of 0.3,

5 ng ml$^{-1}$ aTc dissolved in 100% ethanol was used to induce the cells and an equivalent volume of 100% ethanol was administered to the uninduced cell populations. Following 0 and 7 h, cells were prepared for fluorescent microscopy using the LIVE/DEAD Baclight Bacteria Viability Kit (Thermo Fisher) to characterize the fraction of viable cells across the population. Microscope images were collected using a Zeiss Axio Observer D1 and Plan-Apochromat 63/1.4 Oil Ph3 M27 objective (Zeiss). Cells were imaged using excitation BP 470/40 and emission BP 525/50 (Filter Set 38 High Efficiency) or excitation 560/40 and emission BP 630/75 (Filter Set 45). Images were captured with a Hamamatsu ORCA-Flash4.0 using the ZEN Software (Zeiss). Cell Counter (Fiji)[63] was used to analyse the images and quantify the number of viable and dead cells. Technical replicates consisted of aliquots of *E. coli* cultures that were independently prepared for microscopy using the LIVE/DEAD protocol.

**Statistics.** Statistical analyses and sample sizes for each experiment are described in the figure legends and Methods subsections. Data represent the mean ± 1 s.d., unless noted otherwise. $P \leq 0.05$ was considered significant.

**Code availability.** The authors declare that all computer code supporting the findings of this study is available on request.

**Data availability.** The RNA-seq data in this study have been deposited in the National Center for Biotechnology Information (NCBI) Gene Expression Omnibus (GEO) with accession code GSE94998. All other data supporting the findings of this study are available on request.

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

## Acknowledgements

We would like to thank Karen Lundy for constructing the RNA-seq libraries and Kristala Prather (MIT) for providing an *E. coli* strain for the gluconate measurements. This work was supported by the US Department of Energy (Grant DE-SC0008812) and used the Vincent J. Coates Genomics Sequencing Laboratory at UC Berkeley, supported by NIH S10 Instrumentation Grants S10RR029668 and S10RR027303. O.S.V. was supported by the Simons Foundation at the Life Sciences Research Foundation postdoctoral fellowship.

## Author contributions

O.S.V. and A.P.A. designed the research. O.S.V. and M.T. carried out the experiments. O.S.V. and M.T. performed the computational modelling. O.S.V., M.T. and A.P.A. discussed data analyses and O.S.V. and M.T. performed the analyses. O.S.V., M.T. and A.P.A. wrote the manuscript. S.B. performed gluconate measurements and L.J.G.C. and C.J.P. implemented shotgun proteomics.

## Additional information

**Competing interests:** The authors declare no competing financial interests.

