## [Peer Review File · Nature Communications]

Reviewers' Comments:

Reviewer #1 (Remarks to the Author)

In this manuscript, the authors utilize the MazF ribonuclease toward allocating cellular resources for the expression of synthetic gene networks in *E. coli*. They found that mazF expression halted cellular growth and resulted in the reduction of many cellular mRNAs. In contrast, expressing synthetic mRNAs lacking canonical ACA sequences targeted by MazF resulted in greatly elevated protein levels. The authors showed that this strategy could boost the activity of simple genetic cascades and production of gluconate in a previously engineered strain, and protecting cellular factors involved in translation or RNA decay further boosted synthetic expression. Using a mathematical model, the authors also explored the negative autoregulatory loop formed by MazF targeting its own mRNA. They found that the feedback loop gave more titratable control of MazF levels, particularly as the affinity for its own mRNA increased. Time-series measurements of global mRNA levels following MazF expression further revealed that more ACA sites generally resulted in lower mRNA levels. Three distinct expression patterns were also observed, suggesting distinct responses to the overexpression of this toxin. They finally showed some link between the secondary structure upstream of the target site and mRNA stability, providing some insights into the prediction of targeting strength.

Overall, the work offers an intriguing strategy to allocate resources in synthetic gene circuits by degrading cellular mRNAs but not the synthetic mRNAs. Synthetic biology has been increasingly focused on the interactions between synthetic systems and their cellular environment, such as through balancing metabolic flux between host needs and product yields. This work could become a standard example of how to begin decoupling synthetic and cellular processes toward further improving engineered, cellular devices.

Major comments:

1. The overarching claim is that mazF expression leads to reallocation of cellular resources. However, an alternative explanation is that halted growth can fully explain the boosted protein levels. Rapidly dividing cells would dilute out any expressed reporter protein, whereas this protein would accumulate in halted cells. Furthermore, the RNA-seq data in Figure 5 do not show extensive degradation of the mRNAs at 8 minutes of induction, raising questions about how many resources are freed up. The only supporting evidence is the improved gluconate production, although this could be an unintended effect of expressing a pleiotropic toxin. Given that the article hinges on resource allocation, more evidence is needed to prove this, such as halting growth through other means and showing that the reporter levels are not boosted.
2. mazF is naturally considered a toxin that can kill the cells, which would be a major limitation for some applications. While the authors show that the extent of cell death is only high when the mazF feedback is disrupted, it would also be prudent to show that the halted cells can restore growth once the inducer is removed.

Other comments:

3. The authors work with a mazF-deletion strain. What is the status of the mazE antitoxin? If still present, this could titrate out some of the MazF, affecting the perceived ultrasensitivity of mazF expression and resource allocation.
4. L. 71: "total biomass (OD600) decreased" is incorrect. Instead, at the end point the total biomass was lower for induced cells than for uninduced cells.
5. L. 78: "the expression of mCherry-P degraded..." Expression can't be degraded, but mRNA and protein can.
6. L. 78-88: this effect seems more linked to the onset of stationary phase in the absence of mazF

expression.

7. L. 90-97: it may be difficult to compare mRNA induction given that cell growth is high in one sample and halted in the other.

8. M&M: specify the calculation used to report mCherry measurements. There is some indication on L. 80-81, although this should be specified in the M&M. Was autofluorescence subtracted as part of the calculation?

9. Figure 1b: difficult to differentiate the colors. I would also recommend showing end points along with the kinetic measurements.

10. Figure 2: labeling and figure legend descriptions are insufficient to interpret the results. What are RNase R and EF? How was the MazF ratio measured exactly?

11. More information is needed on any reported biological replicates and how they were conducted.

12. Figure 3: consider adding a sketch of the model and the associate parameters.

13. Figure 3c: how was mazF induction measured, and how is the induction ratio defined?

14. Figure 5b: the label for the horizontal axis is missing.

15. Figure 5b: showing only the average masks the spread of values, which could be considerable. Also, why is there an apparent spike around 10 sites?

16. Figure 5c and S14: what do the n's represent? The number of replicates or the number of target sites?

17. Figure 5c legend: $n = 939$ would imply that 939 biological replicates were conducted.

18. Figure S1: Consider plotting on a square root vertical axis so small values can be visualized. Also consider showing a zoomed-in plot spanning 0 and 10 genes.

Reviewer #2 (Remarks to the Author)

In this paper, the authors show that global manipulation of resource pools through global degradation of endogenous mRNA is a possible way to enhance the performance of a synthetic circuit. Specifically, the proposed method "redirects" translational and posttranslational gene expression resources from endogenous cellular processes to a synthetic circuit through using mazF RNase-enabled degradation of endogenous mRNA in *E. coli*. The contribution of this work is that it shows the feasibility of redirecting important resources, such as ribosomes, to a synthetic gene target while keeping a sufficient amount of viable cells.

Major comments:

The idea is interesting and definitely very timely. There are however two questions that are not addressed in the paper and seem important to assess the utility/feasibility of this approach to increase the amount of a synthetic protein product of interest. The two main questions that arise while reading the paper are as follows:

(1) mcherry experiments (Figure 3): It is known that steady state protein concentration increases in proportion to the inverse of growth rate [1]. The authors' experiments show that growth rate is significantly decreased with the addition of mazF, while also showing that mCherry concentration (fluorescence/OD) is increased with the addition of mazF (Figure 3a and SI Figure 3a). It is unclear if the observed effect is a result of the redirection of resources toward the synthetic plasmid or a result of decreasing dilution rate. When OD decreases, it is not surprising that concentration/OD increases. It seems that another control experiment would be required to assess this. In fact, if the goal of the resource allocation method is to increase the output of a product from a synthetic circuit measured in concentration/OD, this could be accomplished more simply by just decreasing the growth rate. It is unclear if the decrease in growth rate combined with the increase in product concentration/OD that the authors observe results in more product measured in mass. Therefore, from this experiment it is unclear if the redirection approach is useful at all to increase the product mass, which the reviewer understands is the final objective.

(2) gluconate experiments (Figure 2): It is well known that there is a linear relationship between

growth rate and ribosome availability [2, 3]. From Figure 2b, it appears that growth rate was substantially decreased by the re-allocator circuit. If growth rate decreases, the concentration of available ribosomes should decrease [4]. Therefore, even if all the ribosomes become re-allocated to the synthetic genes, there are going to be much fewer. Are you suggesting that even if there are much fewer ribosomes, all being reallocated to the synthetic genes is sufficient to obtain such a large increase in yield? This should be convincingly argued, especially in light of the previous literature cited down here, to demonstrate that these results are in accordance with what is already known. This could be done with mathematical models performing some computation that considers average number of ribosomes allocated to cellular processes and the expected decrease in ribosome concentration due to decreased growth rate.

(3) Missing references that should be cited and compared against in this paper:

1. Klumpp S, Zhang Z, Hwa T (2009) Growth rate-dependent global effects on gene expression in bacteria. *Cell* 139(7):1366–1375
2. Scott M, Gunderson CW, Mateescu EM, Zhang Z, Hwa T (2010) Interdependence of cell growth and gene expression: Origins and consequences. *Science* 330(6007): 1099–1102
3. Y Zhang, "MazF Cleaves Cellular mRNAs Specifically at ACA to Block Protein Synthesis in *Escherichia coli*", *Molecular Cell* (2003)
4. Bremer, H., P. P. Dennis, and M. Ehrenberg, 2003. Free RNA polymerase and modeling global transcription in *Escherichia coli* *Biochimie* 85
5. J. Schifano "tRNA is a new target for cleavage by a MazF toxin" *Nucl. Acids Res.* (2016)

(4) There was very little description of the mathematical model—how it was obtained and what the terms mean. It would be important to include chemical reactions or a paragraph or two describing how it was obtained in the SI and compare with similar existing models. Also, modeling choices and parameters choices should be justified clearly based on data, previous papers, or known molecular mechanisms.

(5) Overall, the paper is well written but could use work with organization. It would be nice to provide an outline of the paper and what was studied in the introduction as well as more distinct subsections. It is difficult to keep track of the larger context with all the specifics the way the paper is organized currently. Also, it should be commented explicitly on the utility of this approach to improve what target circuit function (related to (1) above)

Smaller comments:

1. It is unclear if the protected version of the essential elongation factor (EF-Ts-P) was used in experiments for the remainder of the paper.
2. Missing "of" on line 72
3. Caption for SI Fig 12 (b) and (c) need to be switched.
4. Line 226: Figure 4f, not 4e
5. Units are wrong in SI Table III: reverse binding rate should have units of 1/sec.
6. SI Figure 1 is out of order (is initially referred to between SI Fig 9 and 10 in text)
7. No SI Figure 19b or c exists despite being referred to in text (lines 347, 351).

Reviewer 1:

In this manuscript, the authors utilize the MazF ribonuclease toward allocating cellular resources for the expression of synthetic gene networks in E. coli. They found that mazF expression halted cellular growth and resulted in the reduction of many cellular mRNAs. In contrast, expressing synthetic mRNAs lacking canonical ACA sequences targeted by MazF resulted in greatly elevated protein levels. The authors showed that this strategy could boost the activity of simple genetic cascades and production of gluconate in a previously engineered strain, and protecting cellular factors involved in translation or RNA decay further boosted synthetic expression. Using a mathematical model, the authors also explored the negative autoregulatory loop formed by MazF targeting its own mRNA. They found that the feedback loop gave more titratable control of MazF levels, particularly as the affinity for its own mRNA increased. Time-series measurements of global mRNA levels following MazF expression further revealed that more ACA sites generally resulted in lower mRNA levels. Three distinct expression patterns were also observed, suggesting distinct responses to the overexpression of this toxin. They finally showed some link between the secondary structure upstream of the target site and mRNA stability, providing some insights into the prediction of targeting strength.

Overall, the work offers an intriguing strategy to allocate resources in synthetic gene circuits by degrading cellular mRNAs but not the synthetic mRNAs. Synthetic biology has been increasingly focused on the interactions between synthetic systems and their cellular environment, such as through balancing metabolic flux between host needs and product yields. This work could become a standard example of how to begin decoupling synthetic and cellular processes toward further improving engineered, cellular devices.

Major comments:

1. The overarching claim is that mazF expression leads to reallocation of cellular resources. However, an alternative explanation is that halted growth can fully explain the boosted protein levels. Rapidly dividing cells would dilute out any expressed reporter protein, whereas this protein would accumulate in halted cells.

MazF induction inhibits cell proliferation, which yields lower total biomass at higher MazF concentrations at steady-state (Figure 2b and Supplementary Figure 3a, 10b). By contrast, uninduced cells grow and divide until the population reaches carrying capacity. We show that the **total** fluorescence of mCherry-P (Figure 1b,e) and **production** and **yield** of gluconate (Figure 2c,d) is higher in MazF-induced cells compared to uninduced cells, even though the total biomass is lower in the presence of MazF. In order for MazF-induced cells to yield higher mCherry-P expression and production of gluconate, total synthesis is very likely enhanced.

An alternative explanation is reduction in the levels of the protein degradation machinery, which could increase the concentrations of specific proteins regulated by these proteases. In addition to the lack of evidence that Gdh and mCherry are catalytically degraded in *E. coli*, the unprotected fluorescent reporter (mCherry-U) and gluconate dehydrogenase enzyme (gdh-U) exhibited lower total expression and production of gluconate in the presence of MazF. Potential protein degradation would have the same impact on the protected and unprotected. Together, these data strongly support that MazF alters cellular physiological state to increase the synthesis of protected synthetic circuit genes. Importantly, dividing the fluorescence by OD600 (biomass) does not alter the qualitative relationship, but has a small quantitative effect on the values (see Figure 1e and Supplementary Figs. 2 and 3b).

To illustrate this reasoning quantitatively, we built a computational model to examine the expression of a protected gene (FP) in response to growth inhibition via MazF activity. In the model, we included a logistic growth equation for total cell concentration (N) that was a function of growth rate (λ , see Supplementary Note, Supplementary Table II,III):

$$\frac{dN}{dt} = N \left(\lambda - \frac{N}{K_{cap}} \right), \text{ where } \lambda = k_{trans} \left(\frac{[rq] + [rr]}{P_{tot}} \right)$$

In the first model, the translation rate of FP is constant and thus the synthesis of FP is decoupled from resource competition. In this case, the synthesis rate of FP and growth can be independently modulated. As expected, the cell concentration decreased as a function of the *mazF* transcription rate (α_f), whereas the

concentration of FP per cell increased due to a reduced dilution rate (**Fig. R1a**). The total FP across the cell population ($N \times \text{FP/cell}$) decreased as a function of α_f . In our second model, the translation rate of FP is a function of ribosome concentration (r , Supplementary Note, Supplementary Tables II,III). In this case, the total FP across the cell population *increases* as a function of α_f due to MazF-induced shifts in ribosome partitioning. In sum, our model shows that redistribution of resources in response to MazF is required to yield a higher total protected output expression.

Figure R1. Relationship between *mazF* transcription rate (α_f) and the total steady-state expression of a protected gene for two different models. In the first model, FP is translated at a constant rate and is therefore decoupled from cellular resource competition. In a second model, the translation rate of FP is a function of ribosome concentration. **(a)** Model of decoupled growth and gene expression via constant translation rate of a protected gene (FP). The total steady-state concentration of FP decreases as a function of α_f (right). Total FP is the product of cell concentration (N) and FP per cell. **(b)** Relationship between α_f and steady-state FP concentration for ribosome-dependent translation of FP. In this case, the total steady-state FP across the population increases as a function of α_f due to resource redistribution activity (right).

Furthermore, the RNA-seq data in Figure 5 do not show extensive degradation of the mRNAs at 8 minutes of induction, raising questions about how many resources are freed up. The only supporting evidence is the improved gluconate production, although this could be an unintended effect of expressing a pleiotropic toxin.

According to our transcriptional profiling data, ~65% of transcripts display a decreasing trend in abundance following MazF induction (total number of genes in K1 and K2 clusters in Figure 5c). The number of ribosomes that are made available to protected genes following MazF induction is a function of the ribosome density per transcript and the transcript abundance. Since these parameters are not known, it is difficult to evaluate the total amount of ribosomes that are made available to protected transcripts in response to MazF induction based on the RNA-seq results. However, our data indicated that MazF activity enhances protected gene synthesis rates, suggesting that resources are funneled towards protected genes.

We show that MazF activity enhanced the *total* expression of a protected fluorescent reporter gene and *total* production/yield of gluconate in cells bearing a protected gluconate dehydrogenase (Gdh-P). In addition, an unprotected fluorescent reporter was significantly reduced by MazF activity and cells bearing an unprotected gluconate dehydrogenase (Gdh-U) yielded a lower concentration of gluconate compared to cells expressing Gdh-P (Figure 2c,d). These data, combined with our understanding of the well-characterized functions of MazF, strongly suggest that MazF enhanced the synthesis of protected genes via modifications in cellular physiological state.

Given that the article hinges on resource allocation, more evidence is needed to prove this, such as halting growth through other means and showing that the reporter levels are not boosted.

Based on our knowledge, it is not experimentally feasible to inhibit growth without impacting gene expression. The cellular networks involved in growth and gene expression are highly intertwined and there are numerous mechanisms at play. A potentially feasible strategy would involve physically blocking cell elongation by trapping cells in a chamber approximately the dimension of a single bacterial cell. In addition to the technical difficulties of this experiment (e.g. construction of such device, allowing sufficient and uniform nutrient availability, etc.), this experiment is beyond the scope of our paper.

Due to the difficulty of the proposed experiment, we examined the relationship between gene expression and growth in a computational model. We considered two scenarios: (1) constant translation of a protected gene FP that is decoupled from resource allocation, or (2) ribosome-dependent translation rate of FP, which links the synthesis of FP to limiting ribosome pools. Our results indicate that an alteration in resource partitioning is required to yield an increase in total FP across the population as a function of MazF concentration (**Fig. R1b**).

2. *mazF* is naturally considered a toxin that can kill the cells, which would be a major limitation for some applications. While the authors show that the extent of cell death is only high when the *mazF* feedback is disrupted, it would also be prudent to show that the halted cells can restore growth once the inducer is removed.

Previous studies have characterized the effects of MazE on MazF-induced cell viability (PMID 15576778, 12123459). The authors' results showed that MazF-induced cells can be resuscitated by MazE and the viable fraction of the population depends on the duration of time that cells were exposed to MazF. Even though the impact of MazF on cell viability has been previously studied, we carried out experiments to determine the fraction of MazF-induced cells that can be resuscitated by MazE in our strain background (S3 in Supplementary Table I) and experimental conditions.

In this experiment, *mazF* was regulated by an arabinose-inducible promoter (P_{BAD}), which allowed for transient control of promoter activity via glucose administration (glucose represses the P_{BAD} promoter). Strain S3 (Supplementary Table I) was transformed with a low copy plasmid containing P_{BAD} -*mazF* and a medium copy plasmid containing P_{LAC} -*mazE*. We used a mutant *Lacl* that reduced promoter leakage in the absence of the inducer IPTG (PMID 23834731). Cells were grown to exponential phase and 0.1% arabinose was administered to the cultures. Aliquots of the cultures were removed following 1 and 3 hr of induction and plated on LB plates containing 0 or 1 mM IPTG and supplemented with 1% glucose (**Fig. R2**).

MazF is a stable protein that persists for many hours, whereas *MazE* has a short half-life due to catalytic degradation by the *clpPA* serine protease ($t_{1/2} = 30$ min). As a consequence, while glucose inhibits further synthesis of *MazF*, the accumulated *MazF* proteins cannot be diluted out by cell division or degraded catalytically. Therefore, in the absence of *MazE*, *MazF* activity will continue to impact cellular operations following transcriptional shut down. Consistent with this understanding, our data showed that *MazE* could resuscitate the 100% of the population induced with *MazF* for 1 and 3 hr. By contrast, ~5% and ~0.8% of *MazF*-induced cells exposed to glucose (0 mM IPTG) following 1 and 3 hr remained viable. These data show that *MazE* is a key control parameter that could be used to resuscitate *MazF*-induced cells by rapidly inhibiting *MazF* activity as a function of time.

Figure R2. Exposure of *MazF*-induced cells to *MazE* enables full recovery of the population. Strain S3 (Supplementary Table I) bearing P_{BAD} -*mazF* and P_{LAC} -*mazE* was induced with 0.1% arabinose for 1 and 3 hr and then immediately plated on LB plates containing 1% glucose and 0 or 1 mM IPTG. Plot shows time vs. average colony forming units (CFU) per mL. Error bars represent 1 s.d. ($n = 2$).

3. The authors work with a *mazF*-deletion strain. What is the status of the *mazE* antitoxin? If still present, this could titrate out some of the *MazF*, affecting the perceived ultrasensitivity of *mazF* expression and resource allocation.

We have included a figure (Supplementary Figure 15) of the temporal response of the RNA-seq log₂ fold change of the *mazE* transcript and MazE protein abundance measured by proteomics. These data show a delayed induction of MazE in response to aTc administration, indicating that MazF activity up-regulates MazE levels, forming a negative feedback loop. The MazF protein level is significantly higher than MazE, explaining the lack of inhibition of MazF activity in these conditions.

To examine the role of MazE on circuit properties and growth, we expanded our computational model to include MazE (Supplementary Figure 16). To capture the feedback loop, *mazE* transcription rate (α_e) depends on the concentration of active MazF *mazFpd* (MazF dimer, see Supplementary Note for detailed description of the model). We analyzed the influence of the MazE negative feedback on steady-state MazF levels, growth rate, resource redistribution activity (defined as the translation rate k_{transFP} of a protected reporter gene FP) and ultrasensitivity. Our results highlight that the MazE negative feedback can modulate the parameter regime that maximizes resource redistribution activity and impacts growth rate. While the MazE feedback can enhance ultrasensitivity at high K_{Df} (dissociation constant of MazF to the *mazF* transcript) values in a narrow parameter regime (corresponding to weak MazF mRNA-decay negative feedback), ultrasensitivity decreased across a broad range of α_e values. In sum, MazE is key species in the network that can modulate resource redistribution activity and modulate the impact of MazF on growth. We have included a new results section that includes these results.

4. L. 71: “total biomass (OD600) decreased” is incorrect. Instead, at the end point the total biomass was lower for induced cells than for uninduced cells.

We revised this sentence to state: “The MazF induction ratio of total fluorescence increased (Figure 1e), whereas cells induced with aTc exhibited lower total biomass (Supplementary Figure 3a).”

5. L. 78: “the expression of mCherry-P degraded....” Expression can’t be degraded, but mRNA and protein can.

We revised this sentence to state: “In the presence of aTc, the MazF induction ratio of total fluorescence was enhanced (Fig. 1e), whereas the total biomass was lower (Supplementary Fig. 3a).”

Figure R3. qPCR reference gene *cysG* did not change significantly in response to MazF activity based on RNA-seq. Scatter plot of RPKM measurements prior to induction with MazF vs. induction with MazF (5 ng/ml aTc) for 8 min. The purple data point denotes the *cysG* transcript.

6. L. 78-88: this effect seems more linked to the onset of stationary phase in the absence of *mazF* expression.

We agree with the reviewer that entry into stationary phase contributes to the decrease in fluorescence in uninduced cells. We revised this sentence to state: “These data indicate that heterologous expression was significantly attenuated by delayed induction in the absence of MazF, presumably by the transition from exponential to stationary phase.”

7. L. 90-97: it may be difficult to compare mRNA induction given that cell growth is high in one sample and halted in the other.

In Supplementary Figure 5, we quantified the mCherry-P mRNA fold change by qPCR using the $2^{-\Delta\Delta C_T}$ method. This approach normalized to the geometric mean of two reference genes *rrsA* and *cysG*, and thus accounts for variation in the total RNA across conditions. While the abundance of *rrsA* was not measured (rRNA subtraction was performed prior to construction of the RNA-seq libraries), the *cysG* abundance did not change significantly in our transcriptional profiling data following 8 min of induction (**Fig. R3**).

8. *M&M: specify the calculation used to report mCherry measurements. There is some indication on L. 80-81, although this should be specified in the M&M. Was autofluorescence subtracted as part of the calculation?*

We performed background subtraction for fluorescence and OD600 plate reader measurements. Our data analysis protocol is described in the Materials & Methods section.

9. *Figure 1b: difficult to differentiate the colors. I would also recommend showing end points along with the kinetic measurements.*

We modified the colors and included a bar plot (inset in Fig. 1b) of steady-state mCherry expression.

10. *Figure 2: labeling and figure legend descriptions are insufficient to interpret the results. What are RNase R and EF? How was the MazF ratio measured exactly?*

We updated the Fig. 3 legend to refer to RNase R and EF-Ts and include the definitions of MazF induction and sequence protection ratios. These metrics are also defined in the main text. The MazF induction ratio is defined as the fold change of mCherry-P expression (total fluorescence or divided by OD600) in the presence to absence of MazF. The sequence protection ratio is defined as the ratio of mCherry-P (total fluorescence or divided by OD600) to mCherry-U (total fluorescence or divided by OD600) in the absence or presence of MazF.

RNase R and EF-Ts are key host factors involved in translation and RNA decay. EF-Ts decreased in our proteomics data (Supplementary Fig. 8b), suggesting that protection of this critical translation factor could preserve translational efficiency following MazF induction. We show that protection of these host factors enhanced resource redistribution activity (Figure 3b).

11. *More information is needed on any reported biological replicates and how they were conducted.*

We have included additional information about the replicates for each experimental method in the Materials & Methods section.

12. *Figure 3: consider adding a sketch of the model and the associate parameters.*

We added a figure showing the molecular interactions and key parameters in the model (Supplementary Figure 12).

13. *Figure 3c: how was mazF induction measured, and how is the induction ratio defined?*

The MazF induction ratio is defined as the fold change of mCherry-P expression (total fluorescence or divided by OD600) in the presence to absence of MazF.

14. *Figure 5b: the label for the horizontal axis is missing.*

We added the x-axis label for Fig. 5b.

15. *Figure 5b: showing only the average masks the spread of values, which could be considerable.*

We included a new figure that shows the variation in log₂ fold change of the RNA-seq data as a function of the number of MazF sites (Supplementary Figure 17). Transcript abundance is determined by a balance between the rates of synthesis and decay. As such, transcription rate, activity of RNases beyond MazF, as well as MazF-induced mRNA decay contributes to the overall fold change in transcript abundance. Due to several contributing factors, there is a weak correlation between the number of mazF sites and log₂ fold change at the single transcript level. However, the number of sites is negatively correlated with the average log₂ fold change. Figure 5b highlights the negative correlation between the number of mazF sites and the mean log₂ fold change.

Also, why is there an apparent spike around 10 sites?

The *mazF*, *metA*, *yafE*, *yhdZ* and *dkgB* transcripts contain 9-11 recognition sites and are induced >10-fold following aTc administration.

16. Figure 5c and S14: what do the *n*'s represent? The number of replicates or the number of target sites?

n represents the number of genes in each category. We updated the figure legends to define *n*.

17. Figure 5c legend: *n* = 939 would imply that 939 biological replicates were conducted.

We revised the figure legend to clarify that 939 refers to the number of genes.

18. Figure S1: Consider plotting on a square root vertical axis so small values can be visualized. Also consider showing a zoomed-in plot spanning 0 and 10 genes.

We have updated Figure S1 to show gene length vs. the square root of the number of sites and included a zoomed-in plot (inset) that spans 0-20 genes.

Reviewer #2:

In this paper, the authors show that global manipulation of resource pools through global degradation of endogenous mRNA is a possible way to enhance the performance of a synthetic circuit. Specifically, the proposed method "redirects" translational and posttranslational gene expression resources from endogenous cellular processes to a synthetic circuit through using mazF RNase-enabled degradation of endogenous mRNA in E. coli. The contribution of this work is that it shows the feasibility of redirecting important resources, such as ribosomes, to a synthetic gene target while keeping a sufficient amount of viable cells.

Major comments:

The idea is interesting and definitely very timely. There are however two questions that are not addressed in the paper and seem important to assess the utility/feasibility of this approach to increase the amount of a synthetic protein product of interest. The two main questions that arise while reading the paper are as follows:

(1) mcherry experiments (Figure 3): It is known that steady state protein concentration increases in proportion to the inverse of growth rate [1]. The authors' experiments show that growth rate is significantly decreased with the addition of mazF, while also showing that mCherry concentration (fluorescence/OD) is increased with the addition of mazF (Figure 3a and SI Figure 3a). It is unclear if the observed effect is a result of the redirection of resources toward the synthetic plasmid or a result of decreasing dilution rate. When OD decreases, it is not surprising that concentration/OD increases. It seems that another control experiment would be required to assess this. In fact, if the goal of the resource allocation method is to increase the output of a product from a synthetic circuit measured in concentration/OD, this could be accomplished more simply by just decreasing the growth rate. It is unclear if the decrease in growth rate combined with the increase in product concentration/OD that the authors observe results in more product measured in mass. Therefore, from this experiment it is unclear if the redirection approach is useful at all to increase the product mass, which the reviewer understands is the final objective.

To directly evaluate the effects of MazF activity on resource partitioning would require measuring the ribosome density of the transcriptome in the presence and absence of MazF. However, these experiments are also beyond the scope of our paper. The following key results corroborate that MazF alters resource allocation in cells and the shift in resource partitioning can be co-opted by protected genes.

MazF induction inhibits cell proliferation, which manifests as lower total biomass at higher MazF concentrations at steady-state (Figure 2b and Supplementary Figure 3a, 10b). By contrast, uninduced cells grow and divide

until the population reaches carrying capacity. We show that the **total** fluorescence of mCherry-P (Figure 1b,e) and **production** and **yield** of gluconate (Figure 2c,d) is higher in MazF-induced cells compared to uninduced cells, even though the total biomass is lower in the presence of MazF. In order for MazF-induced cells to yield higher total mCherry-P expression and production of gluconate, total synthesis was very likely enhanced. Importantly, dividing the fluorescence by OD600 (biomass) does not alter the qualitative relationship, but has a small quantitative effect on the values (see Figure 1e, Supplementary Figs. 2 and 3b).

An alternative explanation is reduction in the levels of the protein degradation machinery, which could increase the concentrations of specific proteins that are regulated by proteases. However, an unprotected fluorescent reporter (mCherry-U) and gluconate dehydrogenase enzyme (gdh-U) exhibited lower total expression and production of gluconate in the presence of MazF, respectively. Potential changes in the levels of the protein degradation machinery would have the same impact on the protected and unprotected genes. As Figure 1b,d and Supplementary Figure 2 illustrate, the expression of the unprotected and protected reporters is significantly different in response to MazF. Together, these data strongly support that MazF alters cellular physiological state to increase the synthesis of protected synthetic circuit genes.

To illustrate this reasoning quantitatively, we built a computational model to examine the expression of a protected gene (FP) in response to growth inhibition via MazF activity. In the first model, FP is translated at a constant rate that is decoupled from the other molecular species and hence ribosome levels (**Fig. R1a**). In this case, the synthesis rate of FP and growth can be independently modulated. As expected, the cell concentration (N) decreased as a function of the *mazF* transcription rate (α_f), whereas the concentration of FP per cell increased due to a reduced dilution rate. The total FP across the cell population (N x FP/cell) decreased as a function of α_f .

In the second model, the translation rate of FP is a function of ribosome concentration and thus the synthesis rate of FP is coupled to the translational activity of other species (**Fig. R1b**). In this case, the total FP across the cell population *increases* as a function of α_f due to MazF-induced shifts in ribosome partitioning. In sum, redistribution of resources is required to yield enhance total protected output expression in response to MazF.

Based on our knowledge, it is not experimentally feasible to inhibit growth without impacting gene expression. The cellular networks involved in growth and gene expression are highly intertwined and there are numerous mechanisms at play. A potentially feasible strategy would involve physically blocking cell elongation by trapping cells in a chamber approximately the dimension of a single bacterial cell. In addition to the technical difficulties of this experiment (e.g. construction of such device, allowing sufficient and uniform nutrient availability, etc.), this experiment is beyond the scope of our paper.

(2) gluconate experiments (Figure 2): It is well known that there is a linear relationship between growth rate and ribosome availability [2, 3]. From Figure 2b, it appears that growth rate was substantially decreased by the re-allocator circuit. If growth rate decreases, the concentration of available ribosomes should decrease [4]. Therefore, even if all the ribosomes become re-allocated to the synthetic genes, there are going to be much fewer. Are you suggesting that even if there are much fewer ribosomes, all being reallocated to the synthetic genes is sufficient to obtain such a large increase in yield? This should be convincingly argued, especially in light of the previous literature cited down here, to demonstrate that these results are in accordance with what is already known. This could be done with mathematical models performing some computation that considers average number of ribosomes allocated to cellular processes and the expected decrease in ribosome concentration due to decreased growth rate.

The proportional relationship between growth rate and ribosome availability has been measured in specific conditions (variations in nutrient quality/abundance). The effects of other mechanisms that modulate growth rate on ribosome concentrations, including MazF activity, remain largely unknown. In our experiments, cell growth is inhibited by MazF activity through unknown mechanisms. Therefore, it is difficult to predict the effects of MazF activity on ribosome concentration in cells. On the one hand, MazF inhibits growth rate, which correlates with ribosome concentrations in specific conditions. On the other hand, significant mRNA degradation may increase intracellular resource pools including amino acids, tRNAs and nucleotides, which may stimulate ribosome synthesis. We discuss two results below to address this question.

We used qPCR to quantify the abundance of the protected sfGFP (sfGFP-P) transcript regulated by a ribosomal RNA promoter *rmBP1* (PMID 8071240) or strong synthetic constitutive promoter *apFAB70* (PMID 23474465) on a low copy plasmid in the strain background S2 (Supplementary Table I) in response to MazF induction. Our results showed the sfGFP-P transcript regulated by P_{rmBP1} increased as a function of time in response to 5 ng/ml aTc, which induces MazF activity (**Fig. R3**). By contrast, the fold change of sfGFP-P mRNA regulated by $P_{apFAB70}$ did not significantly change. Although MazF activity induces growth arrest, P_{rmBP1} transcription rate increases as a function of time, illustrating the counterintuitive effect of MazF on promoter activities. Further, a recent study demonstrated that the stringent response is not activated by MazF activity (PMID 25848049), which is consistent with the lack of repression of P_{rmBP1} in response to MazF.

In addition to the promoter activity characterization above, shotgun proteomics measurements of MazF-induced cells demonstrated that the abundance of the majority of detectable ribosomal proteins (r-proteins) did not significantly change over many hours (Supplementary Figure 8a). Together, these results suggest that the concentration of ribosomes does not significantly decrease in MazF-induced cells for a period of time. Future work should perform detailed investigations of the coupling between growth rate inhibition via MazF induction and network activities.

(3) *Missing references that should be cited and compared against in this paper:*

1. Klumpp S, Zhang Z, Hwa T (2009) Growth rate-dependent global effects on gene expression in bacteria. *Cell* 139(7):1366–1375
2. Scott M, Gunderson CW, Mateescu EM, Zhang Z, Hwa T (2010) Interdependence of cell growth and gene expression: Origins and consequences. *Science* 330(6007): 1099–1102
3. Y Zhang, "MazF Cleaves Cellular mRNAs Specifically at ACA to Block Protein Synthesis in *Escherichia coli*", *Molecular Cell* (2003)
4. Bremer, H., P. P. Dennis, and M. Ehrenberg, 2003. Free RNA polymerase and modeling global transcription in *Escherichia coli* *Biochimie* 85
5. J. Schifano "tRNA is a new target for cleavage by a MazF toxin" *Nucl. Acids Res.* (2016)

We added text to the Discussion that cites references 1-2 above. Since the relationship between growth rate modulation via changes in nutrient levels/quality and MazF activity on cellular physiology remains unresolved, a direct comparison of the conclusions of these previous studies and our results is not possible. Corroborating this notion, MazF induction does not induce the stringent response (PMID 25848049), whereas the concentration of ppGpp is inversely related to growth rate for nutrient-limited exponentially growing cells (PMID 26443740)

We cited reference 3-4 in other sections of the manuscript. Reference 5 is not relevant for our study since this paper focuses on MazF-mt9 from *Mycobacterium tuberculosis*, which has different sequence specificity/activities than MazF from *E. coli*. As far as we know, MazF from *E. coli* has not been shown to significantly degrade tRNAs. In fact, the majority of tRNAs are protected from MazF cleavage since they lack 'ACA' recognition sequences.

(4) *There was very little description of the mathematical model—how it was obtained and what the terms mean. It would be important to include chemical reactions or a paragraph or two describing how it was obtained in the SI and compare with similar existing models. Also, modeling choices and parameters choices should be justified clearly based on data, previous papers, or known molecular mechanisms.*

The Supplementary Note and Supplementary Tables II, III provide information about the model including the equations, species and parameter values. We revised the model to include equations to represent the molecular interactions of MazE with other species in the circuit.

(5) Overall, the paper is well written but could use work with organization. It would be nice to provide an outline of the paper and what was studied in the introduction as well as more distinct subsections. It is difficult to keep track of the larger context with all the specifics the way the paper is organized currently.

We revised the final paragraph of the introduction to outline the major results of the paper. We have six major results sub-sections that are delineated in the text by sub-headings: (1) Characterization of inducible MazF; (2) Enhancement of gluconate activity using the MazF resource allocator; (3) Protection of key factors that support synthetic circuit function; (4) Dissecting the role of the MazF mRNA-decay feedback loop; (5) MazE negative feedback loop impacts growth and circuit properties; (6) Time-resolved transcriptional profiling of MazF-induced cells; and (7) Interrogation of parameters that impact MazF activity.

Also, it should be commented explicitly on the utility of this approach to improve what target circuit function (related to (1) above)

We added text to the Discussion section that describes the potential utility of the MazF resource allocator: *"The utility of this approach is to modulate the global physiological state of host-cells to allow focusing of intracellular resources on specific target networks. For example, the MazF resource allocator could be used to enhance target functions that compete directly with biomass synthesis, such as exploiting microbes as "cell factories" to synthesize chemicals or biomolecules of interest. Further, MazF activity could potentially minimize unintended environmental impact due to cell proliferation, while allowing engineered cells to carry out a desired function in a complex environment. Coupling this strategy to dynamic regulation of MazE would enable periodic resuscitation of cellular sub-systems and maintain metabolic activity over longer time scales."*

Smaller comments:

1. It is unclear if the protected version of the essential elongation factor (EF-Ts-P) was used in experiments for the remainder of the paper.

The protected host factors (EF-Ts-P and RNase R-P) were used in Figure 3b and Supplementary Figure 9. To clarify this, we noted which strains were used in each experiment in our results section. Supplementary Table I contains a list of strains and plasmids used in this study.

2. *Missing "of" on line 72*

We corrected this in the manuscript.

3. *Caption for SI Fig 12 (b) and (c) need to be switched.*

We corrected the order of the captions.

4. *Line 226: Figure 4f, not 4e*

We corrected this figure reference.

5. *Units are wrong in SI Table III: reverse binding rate should have units of 1/sec.*

We corrected the units in SI Table III.

6. *SI Figure 1 is out of order (is initially referred to between SI Fig 9 and 10 in text)*

We refer to Supplementary Figure 1a in the beginning of the results section and Supplementary Figure 1b between Supplementary Figures 9 and 10. The plots in Figure 1 both analyze the number of mazF sites across the *E. coli* transcriptome.

7. *No SI Figure 19b or c exists despite being referred to in text (lines 347, 351).*

We corrected this in the manuscript.

Reviewers' Comments:

Reviewer #1 (Remarks to the Author)

The authors have taken sufficient steps to address all comments raised by the reviewers.

Reviewer #2 (Remarks to the Author)

The reviewer thought the authors' rebuttal was convincing, thorough, and well supported and does not have any further comments. Acceptance without revisions of the revised manuscript is recommended.